# Obstructive Sleep Apnoea Severity Is Negatively Associated with Depressive Symptoms: A Cross-Sectional Survey of Outpatients with Suspected Obstructive Sleep Apnoea in Japan

**DOI:** 10.3390/ijerph19095007

**Published:** 2022-04-20

**Authors:** Kazuki Ito, Masahiro Uetsu, Ayaka Ubara, Arichika Matsuda, Yukiyoshi Sumi, Hiroshi Kadotani

**Affiliations:** 1Department of Anesthesiology, Shiga University of Medical Science, Seta-Tsukinowa-Cho, Otsu 520-2192, Japan; sphk96z9@gmail.com; 2Sleep Outpatient Unit for Sleep Apnea Syndrome, Nagahama City Hospital, 313 Ohinui-Cho, Nagahama 526-0043, Japan; mauetsu@yahoo.co.jp; 3Department of Psychiatry, Shiga University of Medical Science, Seta-Tsukinowa-Cho, Otsu 520-2192, Japan; uba.a.1229@gmail.com (A.U.); arichika@belle.shiga-med.ac.jp (A.M.); ysumi@belle.shiga-med.ac.jp (Y.S.); 4Graduate School of Psychology, Doshisha University, Kyoto 610-0394, Japan; 5Japan Society for the Promotion of Science, Research Fellowships, Tokyo 102-0083, Japan

**Keywords:** obstructive sleep apnoea, sleepiness, insomnia, depression, mental health

## Abstract

Background: Multiple clinical departments are involved in the provision of obstructive sleep apnoea (OSA) therapy in Japan. Inconsistent results regarding the association between depression and OSA have been reported. Methods: This cross-sectional survey compared newly diagnosed OSA patients at two outpatient sleep apnoea units in Shiga Prefecture, Japan: one associated with the psychiatry department (*n* = 583), and the other with the otolaryngology department (*n* = 450). Results: The unit associated with the psychiatry department had more patients referred by psychiatrists than that with the otolaryngology department (11% vs. 3% *p* < 0.05). Sleepiness, insomnia, and depression were assessed using the Epworth Sleepiness Scale (ESS), Athens Insomnia Scale (AIS), and Patient Health Questionnaire-9 (PHQ-9), respectively. The ESS, AIS, and PHQ-9 scores were higher in the sleep unit in the psychiatry department (*p* < 0.001 each). Snoring and moderate to severe OSA were more prevalent in the unit attached to the otolaryngology department (*p* < 0.001 each). Patients with moderate to severe OSA had lower PHQ-9 scores than those with no to mild OSA (OR: 0.96, 95% CI: 0.92–1.00, *p* = 0.042). Conclusion: Patients with sleepiness, insomnia, and depressive symptoms were more likely to attend a sleep outpatient unit associated with a psychiatry department, whereas those with snoring and sleep apnoea attended that associated with an otolaryngology department. OSA severity was negatively associated with depressive symptoms.

## 1. Introduction

Obstructive sleep apnoea (OSA) is a common disorder characterized by recurrent episodes of upper airway obstruction during sleep [1,2]. A total of 425 million adults were estimated to have moderate to severe OSA (defined as apnoea–hypopnea index [AHI] ≥ 15 events per h) worldwide [3]. Its prevalence rate was estimated to be 14.0% in Japan [3,4]. Advanced age, male sex, and a higher body mass index (BMI) are associated with an increase in OSA prevalence [1]. OSA is thought to be linked to multiple adverse health outcomes, including daytime sleepiness, decreased quality of life, hypertension, diabetes, coronary artery disease, stroke, atrial fibrillation, congestive heart failure, cognitive function decline, depression, and mortality [5].

Several clinical departments are involved in the provision of OSA therapy in Japan, including psychiatry, otolaryngology, internal medicine, and dentistry [2]. Although multicentre studies focusing on OSA have been conducted in Japan [6,7], the differences in the patients’ characteristics between each clinical department are not well understood.

Depression is the most prevalent mental disorder in Japan [8,9]. Inconsistent results regarding the association between depression and OSA have been reported. Some studies have reported that depressive symptoms are positively associated with OSA severity [10,11] and that unrecognized OSA is highly prevalent in patients with depression [12,13]. Longitudinal studies have suggested an association between OSA and depression [14]. Positive airway pressure (PAP) treatment was reported to improve depression symptoms [5,15]. However, some studies have reported the lack of an association [16,17,18]; moreover, increased OSA severity has been associated with fewer depressive symptoms [19].

OSA screening tools, including the Berlin questionnaire, STOP-BANG questionnaire, STOP questionnaire, four-item screening tool [20], and ESS [21], have been widely used [22]. However, clinical tools, questionnaires, and prediction algorithms are strongly not recommended to diagnose OSA in the absence of polysomnography (PSG) or out-of-centre sleep testing (OCST) [23]. Pulse oximetry, a simple method to obtain physiological signals, may be another useful tool for screening OSA; however, it may be difficult to set an appropriate threshold to distinguish between moderate and severe OSA by oximetry alone [24].

Abnormalities in craniofacial and upper airway anatomy are important risk factors for OSA [25]. The modified Mallampati grade and Friedman tongue position are commonly used scales to assess the oropharynx during visual evaluation [26]. Lateral X-ray cephalograms are widely used to quantify and analyse craniofacial skeletal morphology and the oropharyngeal space [25]. Recently, cone-beam-computed tomography became available to reconstruct and evaluate a three-dimensional image of these structures [25].

One sleep physician was in charge of examining new patients suspected of OSA in two different clinical settings as follows: one was attached to a psychiatry department and the other to an otolaryngology department. He suspected that the severity of OSA and depressive symptoms may differ between these two settings. To test this hypothesis, the same questionnaires were used to assess depressive symptoms and the other two major symptoms of OSA (i.e., sleepiness and insomnia) in these settings.

In this study, we aimed to analyse the following: (1) the differences in the characteristics of patients who attend sleep outpatient units associated with different clinical departments and (2) the association between OSA and depression. We conducted a cross-sectional survey to compare patients newly diagnosed with OSA at two outpatient units for sleep apnoea in Shiga Prefecture, Japan: one associated with a psychiatry department, and the other with an otolaryngology department.

## 2. Materials and Methods

### 2.1. Participants

We collected the medical records of patients referred to two outpatient sleep apnoea units at two public tertiary care centres in Shiga Prefecture in Japan that are located 71 km apart (Figure 1). One unit was attached to the psychiatry department of a university hospital (Shiga University of Medical Science Hospital, Otsu, Japan), which has 612 beds, whereas the other was attached to the otolaryngology department of a city hospital (Nagahama City Hospital, Nagahama, Japan) that has 600 beds. 

The Shiga University of Medical Science is the only university hospital in Shiga Prefecture and is located on the border of Otsu City (344,000 inhabitants, the largest city and prefectural capital of the Shiga Prefecture) and Kusatsu City (136,000 inhabitants, the second-largest city in the prefecture). The university hospital has a sleep centre that is the only one certified by the Japanese Society of Sleep Research (JSSR) in Shiga Prefecture, with five board-certified physicians of the JSSR. Only one physician is in charge of examining new patients suspected of OSA in the university hospital.

On the other hand, the Nagahama City Hospital is the only hospital with a sleep specialist on staff (part-time) in Nagahama City (116,000 inhabitants, the third-largest city in the prefecture). The physician in charge of new patients with OSA at the university hospital is also in charge of new patients with OSA at the city hospital, who is experienced in the diagnosis of mental disorders in a non-psychiatric patient population [8]. Patients with medical referral letters are given priority to make reservations for appointments in the hospitals, but patients can also make appointments directly without referral letters by paying extra fees.

We retrospectively analysed the medical records of consecutive outpatients who visited one of two hospitals for the first time from 1 April 2015 to 31 March 2019. In this survey, we used PSG or OCST records. We excluded patients who cancelled PSG or OCST and those with no or incomplete PSG/OCST data (Figure 2). We also excluded patients aged < 15 years and those with dementia and intellectual developmental disorders. The medical records also provided information on whether the patients were referred to the hospitals or the patients made the appointments directly by themselves.

We estimated the approximate distance between the residential area of each patient and their hospital using Google Maps ver. 3.46 (Google LLC, Mountain View, CA, USA).

### 2.2. Questionnaires

Sleepiness and insomnia are common symptoms in patients with OSA [1]. Insomnia and depression are highly prevalent and frequently co-occur [27]. Sleepiness, insomnia, and depression were assessed using the Japanese versions of the Epworth Sleepiness Scale (ESS) [21], Athens Insomnia Scale (AIS) [28], and Patient Health Questionnaire-9 (PHQ-9) [29], respectively. To assess and compare these symptoms, we used a set of questionnaires, including ESS, AIS, and PHQ-9, in both clinical and epidemiological settings. We used these questionnaires for patients at the time of a new visit and twice a year thereafter both in the university hospital [30] and in the city hospital 24]. We also used these questionnaires annually in the cohort study performed in Koka City in Shiga Prefecture [27,31].

The ESS is a self-administered questionnaire with eight questions evaluating daytime sleepiness. The respondents are asked to rate their usual chances of dozing off or falling asleep while engaged in eight different activities [21]. A sum score is calculated (range: 0–24), with higher scores indicating more sleepiness.

The AIS was developed by the World Health Organization based on the criteria of the International Classification of Disease, 10th revision. The AIS comprises eight questions, including five assessing nocturnal sleep problems and three evaluating the daytime consequences of insomnia [28]. A sum score is calculated (range: 0–24), with higher scores indicating more severe insomnia symptoms.

The PHQ-9 comprises nine items derived from the Diagnostic and Statistical Manual of Mental Disorders-IV (DSM-IV) diagnostic criteria for depressive disorder, and it is considered a valid instrument for the assessment of depressive symptoms [29]. A sum score is calculated (range: 0–27), with higher scores indicating more severe depression.

### 2.3. OSA Diagnosis

In the Japanese health insurance system, continuous positive airway pressure (CPAP) therapy, the first-line treatment for OSA [2,5], can be prescribed only when patients have respiratory event indexes (REI) ≥ 40 with OCST or AHI ≥ 20 with PSG [2]. PSG tests are strongly recommended to be performed when patients are highly suspected to have OSA after OCST and with REI < 40 [2].

When patients were referred to the Shiga University of Medical Science or the Nagahama City Hospital for OSA diagnosis/treatment, or when patients directly came to the hospital without medical referral letters, all OSA-suspected patients were asked to undergo OCST/PSG. The respiratory event indexes (REIs) obtained from OCST data were manually analysed using Morpheus (Teijin, Tokyo, Japan) or PulSleep LS120 (Fukuda Denshi, Tokyo, Japan) in the city hospital or Smartwatch PMP-300E (Pacific Medico Co., Ltd., Tokyo, Japan) in the university hospital. OCST data were analysed one to two weeks after the patients returned the OSCT devices to the hospitals. The OCST results in the university hospital were manually analysed by technicians blind to the questionnaire results and were reviewed by a board-certified physician of the JSSR, whereas those in the city hospital were manually analysed by the same physician without referring to the questionnaire results. The in-laboratory PSG results were analysed with Alice 5 (Philips Respironics, Inc., Murrysville, PA, USA) blind to the questionnaire results in both settings. We followed the recommended American Academy of Sleep Medicine (version 2.3) scoring criteria [32]. Apnoea was defined as the cessation of airflow for at least 10 s, while hypopnea was defined as a reduction in the airflow amplitude or respiratory effort of at least 30%, with an oxygen desaturation value of 3% or greater for at least 10 s. The numbers of patients diagnosed using PSG and OCST were 250 and 333 in the university hospital and 153 and 297 in the city hospital, respectively. OSA severity was identified based on the recorded AHI or REI; AHI/REI < 5, 5 ≤ AHI/REI < 15, 15 ≤ AHI/REI < 15, and 30 ≥ AHI/REI were classified as no, mild, moderate, and severe OSA, respectively. We divided all AHI/REI scores into two groups: AHI/REI < 15 (none-to-mild OSA) and AHI/REI ≥ 15 (moderate-to-severe OSA) [23].

### 2.4. Statistical Analysis

Descriptive statistics for clinical characteristics are expressed as the mean ± standard deviation (SD) or number (%). Independent *t*-tests and chi-squared (χ^2^) tests were used for continuous and categorical variables, respectively. Bonferroni correction was used for multiple comparisons.

We performed a logistic regression analysis with the hospital choice (0: the city hospital; 1: the university hospital) as the dependent variable after adjusting for age, sex, BMI, snoring, ESS, AIS, PHQ-9, AHI/REI, hypertension, distance from the university hospital, and distance from the city hospital. We also performed a logistic regression analysis with moderate to severe OSA as the dependent variable and adjusted for age, sex, BMI, snoring, ESS, AIS, PHQ-9, lifestyle-related diseases (diabetes, hypertension, cardiovascular diseases, and cerebrovascular diseases), and mental disorders. The Kruskal–Wallis test was used to compare the PHQ-9 scores and OSA severity for sensitivity analysis. The significance level was set to *p* < 0.05. All data were analysed using the SPSS 25.0 statistical software (SPSS Inc., Chicago, IL, USA) and MedCalc ver. 20.014 (MedCalc Software Ltd., Ostend, Belgium). 

In a previous study [8], we could not detect a difference in OSA between controls and participants with depression, and attributed this finding to the small number of participants with depression. We reported an AHI/REI difference between controls and participants with depression of 2.24 (SD: 11.4). In this study, the ratio of participants with PHQ-9 ≥ 10 and those with PHQ-9 < 10 was approximately 0.9. If the true difference in the experimental and control means is 2.24, we needed to study 388 participants with depression and 432 control participants to be able to reject the null hypothesis that the population means of the experimental and control groups are equal with a probability (power) of 0.8. The type-I error probability associated with this test of this null hypothesis is 0.05. Thus, we may have a sufficient sample size to detect an association between OSA and depression in this study.

## 3. Results

The total number of new outpatients was 1156. Overall, 681 participants visited the university hospital and 475 visited the city hospital. After applying the exclusion criteria, the final number of participants included in this study was 583 in the university hospital and 450 in the city hospital (Figure 2).

The proportion of patients referred from other hospitals to the university hospital was significantly higher than that referred to the city hospital (17% vs. 5%, *p* < 0.05; Figure 3). The proportion of patients referred from clinics to the university hospital was significantly lower than that referred to the city hospital (45% vs. 59%, *p* < 0.05).

The proportion of referrals from the psychiatry department of hospitals and clinics was significantly higher in the university hospital than in the city hospital (11% vs. 3%, *p* < 0.05) (Figure 4). No significant differences were found in the other departments, including the otolaryngology department (7% vs. 10%). Seven, twenty-seven, and thirty-one patients were referred from the psychiatry department of the same university hospital, other hospitals, and other clinics to the OSA outpatient unit of the university hospital, respectively. A total of 14, 1, and 29 patients were referred from the otolaryngology department of the same city hospital, other hospitals, and other clinics to the OSA outpatient unit of the city hospital, respectively.

Most patients showed symptoms of OSA, such as snoring, and experienced apnoea during sleep (89.0% in the university hospital and 95.5% in the city hospital). The number of new outpatients from Shiga Prefecture was 603 in the university hospital and 470 in the city hospital. There were 74 and 5 patients from outside Shiga Prefecture in the university hospital and city hospital, respectively (Figure 1). 

Age and AHI/REI were significantly higher in patients visiting the city hospital than in those visiting the university hospital. The prevalence of snoring, moderate to severe OSA, and hypertension was also significantly higher among patients visiting the city hospital. In contrast, the ESS, AIS, and PHQ-9 scores were significantly higher in patients in the university hospital than in those in the city hospital. In addition, the prevalence of mental disorders was significantly higher and the distance from their residence to their hospital was significantly greater among the university hospital outpatients (Table 1). 

We classified patients into two groups according to the severity of OSA (AHI/REI < 15 vs. AHI/REI ≥ 15). Regardless of OSA severity, the AIS scores were significantly higher in the university hospital, while the rate of snoring was significantly higher in the city hospital. The prevalence of lifestyle-related diseases and mental disorders was not significantly different between settings within each OSA severity category (Table 2).

Age, insomnia (AIS), and distance from the hospitals were associated with the hospital choice (Table 3). Depressive symptoms (PHQ-9) and REI/REI were associated with the choices in the unadjusted model, but not in the adjustment model.

The PHQ-9 scores were lower in patients with moderate to severe OSA than in those with no to mild OSA (Figure 5). This association between the PHQ-9 score and OSA severity was also found in the logistic regression analysis, even after adjusting for age, sex, BMI, snoring, ESS, AIS, AHI/REI, hypertension, distance from the university hospital, and distance from the city hospital (Appendix A).

## 4. Discussion

We compared the characteristics of outpatients newly diagnosed with OSA who visited two sleep apnoea units associated with different departments in Japan. In both units, the chief complaints of new outpatients were snoring and witnessed apnoea: 89.0% in the university hospital and 95.5% in the city hospital.

We conducted this study using the same evaluation criteria for two hospitals of the same scale in Shiga Prefecture and found significant differences in age, sleepiness scores, insomnia scores, depression scores, prevalence of mental disorders, snoring rate, and the presence of hypertension. These results indicate that the characteristics of patients referred to sleep units may be affected by the department they are associated with.

A comparison of outpatients between the two sleep apnoea units revealed significantly different patient characteristics. More outpatients with snoring were found in the city hospital, which is associated with the otolaryngology department, while more outpatients with insomnia symptoms were found in the university hospital, which is associated with the psychiatry department. General practitioners must decide which departments are suitable for their outpatient’s suspected OSA; at that time, they may also decide where to refer their outpatients based on other symptoms, such as snoring or witnessed apnoea. 

Physicians in hospitals may have a tendency to refer patients more to the university hospital than to the city hospital, whereas those in clinics tend to refer them to the city hospital. Our findings suggest that distance from home to the hospitals is a main contributing factor to deciding which hospital to attend. Patients tended to be referred to university hospitals from more distant locations. More OSA patients with insomnia were referred to the sleep outpatient unit attached to the psychiatric department in the university hospital. Future studies comparing more hospitals are required to clarify whether these choices are attributed to the characteristics of the hospitals (university hospital vs. city hospital) or related to the department (otolaryngology vs. psychiatry).

Otolaryngologists are recognized as specialists in treating snoring [33]. Uvulopalatopharyngoplasty (UPPP) was first introduced by a Japanese otolaryngologist for snoring and OSA [34]. In a meta-analysis, the success rate of UPPP for OSA was reported to be 51.5% [35]. The serious nonfatal complication rate of UPPP was 1.5%, and its 30-day mortality rate was 0.2% [36]. Thus, UPPP is now considered as a secondary or optional therapy for OSA when patients are intolerant to CPAP or oral appliances [2]. Thus, we assume that general practitioners would tend to refer patients who are strongly suspected to have OSA due to snoring or witnessed apnoea to a nearby unit associated with an otolaryngology department. The proportion of patients with snoring and moderate-to-severe OSA was higher in the city hospital, which is associated with the otolaryngology department (Table 1). However, the adjusted logistic regression analysis results suggest that snoring and OSA may not be the main factors affecting the hospital choice (Table 3). In addition, the proportion of patients referred by otolaryngologists was similar in both hospitals (Figure 4). 

On the other hand, the patients referred to the outpatient sleep apnoea unit in the university hospital had significantly higher AIS, ESS, and PHQ-9 scores than those in the city hospital (Table 1). The AIS, ESS, and PHQ-9 scores of the patients who visited the city hospital were similar to those of the city government employees in Shiga Prefecture, as reported in a previous epidemiological study, where the AIS, ESS, and PHQ-9 scores were 4.98 ± 3.57, 7.85 ± 4.54, and 4.65 ± 4.54, respectively [27]. 

The prevalence of mental illness was significantly higher in the university hospital (21.3%) than in the city hospital (13.8%). Mental disorders, including depression, are strongly associated with insomnia [27], and a psychiatrist mainly treats insomnia, i.e., the university’s outpatient unit for sleep apnoea is attached to the psychiatry department. If the patients were suspected to have any psychiatric disorders in addition to OSA, a general practitioner might hesitate to refer them to the nearby sleep units associated with the otolaryngology department, regardless of OSA severity. The logistic regression analysis results suggested that insomnia symptoms may be an important factor to determine which hospital to attend (Table 3). Moreover, the proportion of patients referred by psychiatrists was higher in the university hospital than in the city hospital (Figure 4). Hospital distance (the distance from the patients’ residence to their hospital) was significantly greater among patients attending the university hospital than among those attending the city hospital. Although there are several sleep units associated with otolaryngology departments in the Shiga Prefecture, the sleep unit in the university hospital was the only one associated with a psychiatry department. This may explain why there was a greater distance to the hospital in patients attending the university hospital and why patients attending the city hospital had a higher mean AHI/REI score and a higher moderate-to-severe OSA prevalence.

Our results suggest a significant relationship between moderate to severe OSA and older age, male sex, BMI, and hypertension, similar to the results of a previous study (Appendix A) [1]. In contrast, no significant association was found between moderate to severe OSA and lifestyle-related diseases, other than hypertension. After PAP treatment, blood pressure was reported to be significantly reduced, but cardiovascular events and glycaemia were not [5]. The triacylglycerol and total cholesterol levels have been reported to improve after PAP treatment; however, the changes attributable to PAP were less significant than those explained by circadian changes [37]. In this study, we only detected an association with hypertension, probably because of the small effects of OSA on lipidaemia and glycaemia and the too-small sample size to detect cardiovascular or cerebrovascular events.

We found that OSA is negatively associated with depressive symptoms (PHQ-9 score) (Appendix A, Figure 5). Some previous studies have reported that depressive symptoms are positively associated with OSA severity [10,11], whereas other studies found no association between depression and OSA [16,17,18]. A recent systematic review and meta-analysis concluded that there was no compelling evidence of an association between OSA and depression in six cross-sectional studies [14]. A study conducted in a large clinical setting in Norway that analysed PSG data (*n* = 3770) reported that the prevalence of depressive symptoms significantly decreases as OSA severity increases [19]. Thus, based on cross-sectional studies, there may be no or a negative association between OSA severity and depression symptoms. Patients with depression were more likely to report somatic symptoms [38] and, therefore, be more sensitive to and overreport their physical symptoms related to OSA. Moreover, insomnia may be involved as a confounder for OSA severity. In other words, if insomnia is the main complaint, but the patient is examined with suspicion for OSA, insomnia and depression could be negatively associated with OSA severity, if there are many cases of insomnia and no to mild OSA. However, this possibility is unlikely, since depression and OSA were negatively correlated in both the university and the city hospitals, and the insomnia (AIS), sleepiness (ESS), and depression (PHQ-9) scores in the city hospital were comparable to those of the municipal employees in the same prefecture [27]. Cognitive function is impaired by OSA [39]. OSA might reduce self-awareness and the perception of stress, which might help reduce depressive symptoms in OSA patients. However, since OSA does not affect the global cognition domain of cognitive functions [39], it may be unlikely to help reduce depressive symptoms.

This study had some limitations. First, it was performed in clinical settings within Shiga Prefecture in Japan and may, therefore, not be representative of the Japanese general population. Second, as only two hospitals were compared, it is impossible to determine which is more important, the nature of the hospital (university hospital or city hospital) or the department attached to the outpatient sleep apnoea units. Third, owing to the cross-sectional study design, longitudinal changes were not analysed. Other limitations include the fact that depressive symptoms were assessed using PHQ-9 only and not by structured interviews. However, in our previous cohort study of a working population in the Shiga Prefecture (NinJa Sleep Study: Night in Japan Home Sleep Monitoring Study) [27,31], the same questionnaires were used. We plan to compare the association between OSA and depression found in the previous study and this study. We used the same in-laboratory PSG device; however, the OCST devices were different between the two settings. A comparison between AHI from PSG and REI from OCST was previously performed in the city hospital, which suggested a reasonable validity of REI compared with AHI [24]. The same sleep physician reviewed all of the analysed results; however, the PSG/OCST analyses were performed by different staff in both settings.

## 5. Conclusions

Patients with sleepiness, insomnia, and depressive symptoms were more likely to attend a sleep outpatient unit associated with a psychiatry department, whereas those with snoring and sleep apnoea tended to attend one associated with an otolaryngology department. OSA severity was negatively associated with depressive symptoms.

## Figures and Tables

**Figure 1 ijerph-19-05007-f001:**
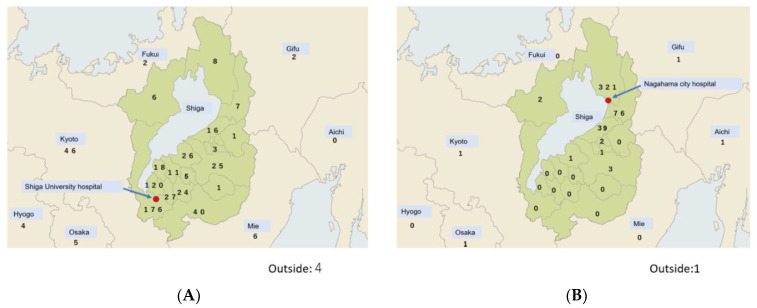
Distribution of new patients in the two settings studied. The numbers of new patients who visited each sleep outpatient unit from 1 April 2015 to 31 March 2019 at Shiga University of Medical Science Hospital (**A**) and Nagahama City Hospital (**B**) are indicated. The green areas represent Shiga Prefecture. Within Shiga Prefecture, each city or county is indicated separately. Prefectures outside Shiga Prefecture are also presented separately. The red circles indicate the locations of the two hospitals.

**Figure 2 ijerph-19-05007-f002:**
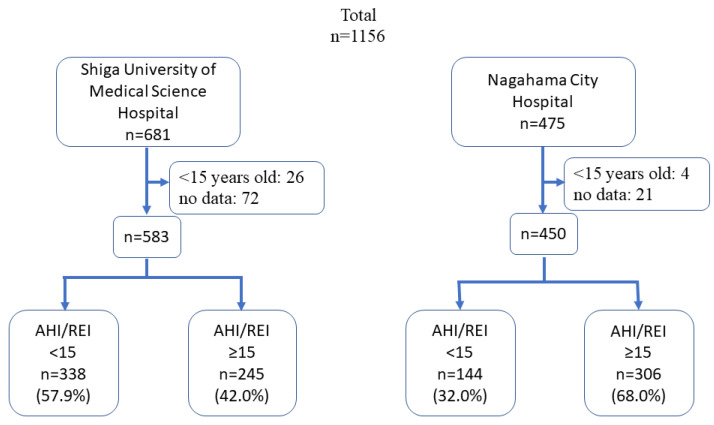
Flowchart of patient inclusion. AHI, apnoea–hypopnea index; REI, respiratory event index.

**Figure 3 ijerph-19-05007-f003:**
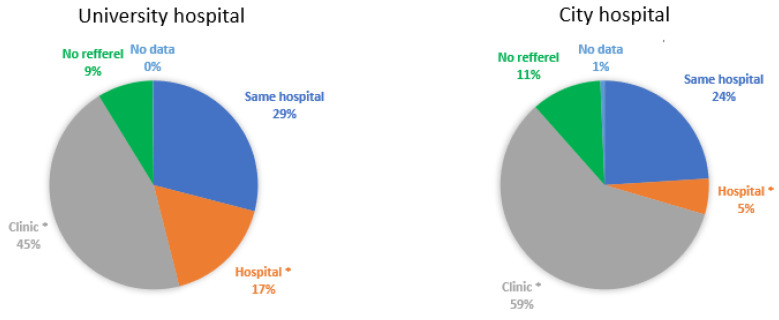
Referral source. Patients were referred from the same hospital (blue), other hospitals (orange), or other clinics (grey). Patients could directly come to the outpatient units without medical referral letters (green). Data regarding the referral sources were missing in some patients (light blue). ^‡^
*p* < 0.05 between the university hospital and the city hospital with Bonferroni correction.

**Figure 4 ijerph-19-05007-f004:**
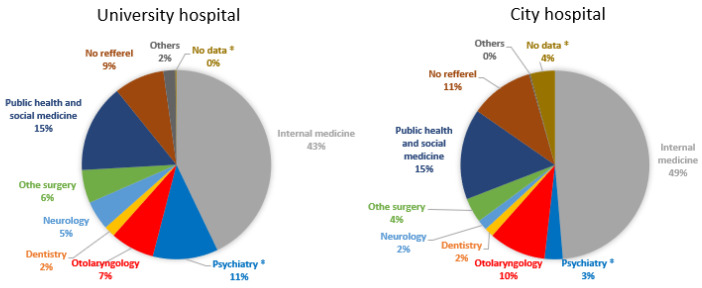
Referral source of OSA-suspected patients. Referrals from OSA-related departments, including internal medicine (grey), psychiatry (blue), otolaryngology (red), dentistry (orange), and other departments, including neurology (light blue) and others (grey), are shown. Surgical departments were separated into otolaryngology (red) and other surgical departments (light blue). Industrial physicians and public health departments were combined into public health and social medicine (navy blue). Some patients had no data regarding the referral source or referral departments (ochre). Patients could directly come to the outpatient units without medical referral letters (brown). ^‡^
*p* < 0.05 between the university hospital and the city hospital with Bonferroni correction.

**Figure 5 ijerph-19-05007-f005:**
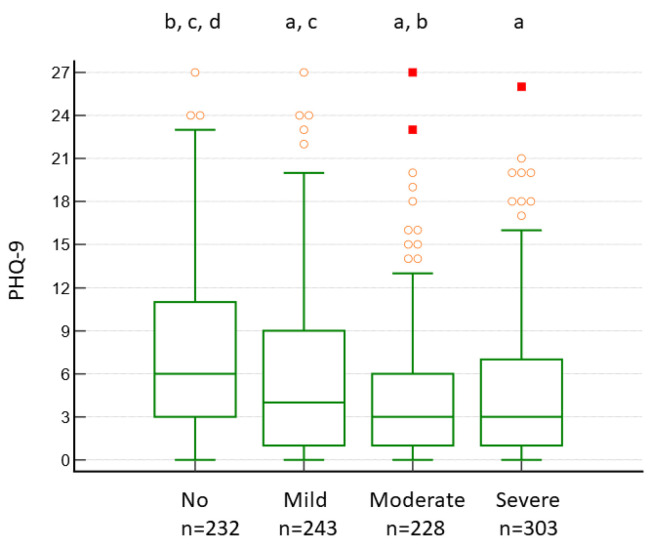
Box-and-whisker plot for PHQ-9 scores according to OSA severity. Patients with AHI/REI < 5, 5 ≤ AHI/REI < 15, 15 ≤ AHI/REI < 15, and 30 ≥ AHI/REI were classified as having no, mild, moderate, and severe OSA, respectively. OSA, obstructive sleep apnoea; PHQ-9, Patient Health Questionnaire-9; AHI, apnoea–hypopnea index; REI, respiratory event index; ■, far outside values; ○, outside values; a, *p* < 0.05 vs. no; b, *p* < 0.05 vs. mild; c, *p* < 0.05 vs. moderate; d, *p* < 0.05 vs. severe.

**Table 1 ijerph-19-05007-t001:** Characteristics of patients included in the study (*n* = 1033).

	Total	University Hospital (*n* = 583)	City Hospital (*n* = 450)	*p*
Age (years)	54.6 ± 16.8	51.3 ± 17.5	58.8 ± 14.8	<0.0001
Male (%)	71.8	68.8	75.8	0.013
Snoring (%)	62.8	57.8	69.3	<0.0001
BMI (kg/m^2^)	26.1 ± 5.38	26.1 ± 5.42	26.3 ± 5.33	0.557
ESS	9.17 ± 5.80	10.0 ± 6.06	8.09 ± 5.26	<0.001
AIS	5.72 ± 4.32	6.53 ± 4.25	4.71 ± 4.21	<0.001
PHQ-9	5.49 ± 5.22	6.27 ± 5.32	4.53 ± 4.92	<0.001
AHI/REI (per h)	23.1 ± 22.1	19.6 ± 22.3	27.6 ± 20.9	<0.001
Moderate to severe OSA (%)	53.3	42.0	68.0	<0.001
Hypertension (%)	44.1	39.5	50.0	0.001
Diabetes (%)	19.5	19.0	20.0	0.699
Hyperlipidaemia (%)	27.0	26.4	27.8	0.625
Cardiovascular diseases (%)	10.2	8.75	12.0	0.086
Cerebrovascular diseases (%)	5.71	6.17	5.11	0.883
Mental disorders (%)	18.0	21.3	13.8	0.002
Hospital distance (km)	17.2 ± 21.4	22.6 ± 24.7	10.3 ± 13.2	<0.001

Bonferroni-corrected *p*-value for 0.05 is 0.003 (= 0.05/16). Data are expressed as the mean ± standard deviation or number of participants (%). Hospital distance, the distance between patients’ residence and their hospital; OSA, obstructive sleep apnoea; BMI, body mass index; PHQ-9, Patient Health Questionnaire-9; AIS, Athens Insomnia Scale; ESS, Epworth Sleepiness Scale; AHI, apnoea–hypopnea index; REI, respiratory event index.

**Table 2 ijerph-19-05007-t002:** Characteristics of patients with no to mild OSA (AHI/REI < 15) vs. moderate to severe OSA (AHI/REI ≥ 15) in the two settings.

	AHI/REI < 15			AHI/REI ≥ 15		
	University Hospital (*n* = 338)	City Hospital (*n* = 144)	*p*	University Hospital (*n* = 245)	City Hospital (*n* = 306)	*p*
Age (years)	47.7 ± 18.6	54.0 ± 16.4	0.001	56.3 ± 14.5	61.1 ± 13.4	<0.001
Male (%)	61.5	66.7	0.286	78.8	80.1	0.709
BMI (kg/m^2^)	24.5 ± 4.62	25.4 ± 5.09	0.059	28.2 ± 5.69	26.7 ± 5.40	0.001
Snoring	55.5	75.7	<0.001	60.8	66.3	<0.001
ESS	10.7 ± 6.36	8.46 ± 5.66	<0.001	9.11 ± 5.50	7.91 ± 5.07	0.009
AIS	6.86 ± 4.39	5.17 ± 4.49	<0.001	6.04 ± 3.99	4.49 ± 4.05	<0.001
PHQ-9	6.80 ± 5.60	6.00 ± 5.83	0.163	5.47 ± 4.79	3.84 ± 4.26	<0.001
AHI/REI (per h)	4.93 ± 4.60	7.93 ± 4.36	<0.001	39.8 ± 21.2	36.9 ± 19.1	0.940
Hypertension	28.7	33.3	0.310	54.3	58.2	0.361
Diabetes	14.8	14.6	0.953	24.9	22.5	0.519
Hyperlipidaemia	22.8	22.2	0.893	31.4	30.4	0.794
Cardiovascular diseases	7.10	9.03	0.467	11.0	13.4	0.399
Cerebrovascular diseases	2.66	4.86	0.218	8.98	5.23	0.084
Mental disorders	25.1	21.5	0.395	15.9	10.1	0.043

Bonferroni-corrected *p*-value for 0.05 is 0.003 (= 0.05/14). Data are expressed as the mean ± standard deviation or number of participants (%). OSA, obstructive sleep apnoea; BMI, body mass index; PHQ-9, Patient Health Questionnaire-9; AIS, Athens Insomnia Scale; ESS, Epworth Sleepiness Scale; AHI, apnoea–hypopnea index.

**Table 3 ijerph-19-05007-t003:** Logistic regression analysis: comparisons between the university hospital and the city hospital.

	Unadjusted Odds Ratio (95% CI)	*p*	Adjusted Odds Ratio (95% CI)	*p*
Age (years)	0.97 (0.97–0.98)	<0.001	0.98 (0.95–0.99)	0.034
Male	0.71 (0.54–0.94)	0.015	0.72 (0.35–1.48)	0.365
BMI (kg cm^−2^)	0.58 (0.97–1.02)	0.581	0.97 (0.90–1.04)	0.411
Snoring	0.61 (0.47–0.79)	<0.001	0.74 (0.38–1.46)	0.389
ESS	1.06 (1.04–1.09)	<0.001	1.02 (0.96–1.09)	0.466
AIS	1.11 (1.08–1.15)	<0.001	1.14 (1.03–1.26)	0.008
PHQ-9	1.07 (1.04–1.10)	<0.001	0.95 (0.87–1.03)	0.221
AHI/REI (per h)	0.98 (0.98–0.99)	<0.001	0.99 (0.97–1.00)	0.080
Hypertension	0.66 (0.51–0.84)	0.001	0.84 (0.39–1.80)	0.660
Distance from the university hospital (km)	0.90 (0.86–0.91)	<0.001	0.92 (0.90–0.94)	<0.001
Distance from the city hospital (km)	1.13 (1.11–1.15)	<0.001	1.07 (1.06–1.09)	<0.001

BMI, body mass index; PHQ-9, Patient Health Questionnaire-9; AIS, Athens Insomnia Scale; ESS, Epworth Sleepiness Scale; AHI, apnoea–hypopnea index; REI, respiratory event index.

## Data Availability

The datasets generated during and/or analysed during the current study are available from the corresponding author on reasonable request.

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
