# Peer review of "Obstructive Sleep Apnoea Severity Is Negatively Associated with Depressive Symptoms: A Cross-Sectional Survey of Outpatients with Suspected Obstructive Sleep Apnoea in Japan"

_ijerph, 2022, doi:10.3390/ijerph19095007_

Round 1

Reviewer 1 Report

Interesting idea but lacks robused scientific proving and there are some methological issues taht need to be resolved

Language and organization is fine but when it comes to comparison between two different hospitals and how they include patients for OSA testing and the interpentation of the OSA scoring by two methds PSG and HSAT. Also who did the referals to the hospitals. HOw blinded they were?

Issues that need an answer since the study is at the very center of that difference that they tried to understant  it

Author Response

Thank you very much for your comments. We have revised our manuscript to address all the issues you pointed out. Please see the attachment.

Reviewer 2 Report

The introduction would be strengthened by adding a paragraph explaining why the authors believe there may be a difference in the characteristics of patients who attend sleep outpatient units associated with different clinical departments. Has any data suggested that departments take different measures or that patients with different characteristics choose to go to one outpatient unit versus another?

Thank you for including a detailed power analysis in the Methods section.

Table 3 shows associations between AHI score and patient features, but does not add any new information to the context of this manuscript. While it is important to evaluate associations between AHI and patient features, I do not think this table strengthens the manuscript.

Figure 2 is not convincing. Perhaps depicting the data in a different format would highlight any significant differences between groups. 

The Discussion and Conclusion sections posit that individuals with greater depression or insomnia symptoms are being sent to a psychiatry-associated sleep unit over a sleep unit associated with an otolaryngology department. This hypothesis, or similar reasoning for conducting this study, should be included in the introduction. 

The data presented in the Results section does not strongly suggest that patients with different symptoms are being sent to different hospital units. Additional statistical tests, such as a binomial logistic regression using patient features to predict whether they had gone to the university vs. city hospital, or a discriminant function analysis, may be able to provide a clearer picture. 

I applaud the authors for estimating distance between patients' homes and the hospitals. This is an inventive way to examine logistical reasons for going to one hospital versus another.  However, this analysis is under-utilized. It would be very persuasive to see a figure or map examining the distance between the university sleep unit and patients' homes compared to the city hospital or an alternative sleep unit that would be closer to patients' home but not associated with a psychiatry wing.  Currently, there is not enough information about general practitioners' reasoning or patients' reasoning when selecting a sleep unit to support the idea that patients with greater depressive symptoms are being sent to a particular unit. This is a limitation, and should be addressed in the manuscript. 

I recommend adding more targeted statistical tests and replacing Figure 2 and Table 3 with clearer tables or figures that strongly communicate the most salient findings. 

Author Response

(The authors gave the same response as above.)

Reviewer 3 Report

This is an interesting study about OSA severity and its' inverse association with depressive symptoms - cross-sectional survey was performed by authors in Japanese outpatients with suspected OSA. While entire article is very well planned and conducted, some criticism should be raised prior to publication. See below:

Title is miselading and inconsistent with latter sections: ' Obstructive sleep apnoea severity is inversely associated with depressive symptoms: a cross-sectional survey of outpatients with suspected obstructive sleep apnoea in Japan' while later in Abstract section authors claim that ' OSA severity was negatively associated with depressive symptoms' - inverse assocaition and negative association are not synonyms and at least one of those statements should be re-formulated for clarity.

Abstract: This section should be structurized according to common MDPI publication policy. Please see https://www.mdpi.com/journal/ijerph/instructions for details

Introduction

L38 and 45-48 authors start diagnosing OSAS with PSG, however no other screening tools were mentioned, as approx. one out of four snoring patients has OSAS. I think authors should elaborate briefly on these: CBCT, Mallampati, Berlin, Stop-Bang questionnaire as most up-to-date screening tools; please incorporate and cite https://pubmed.ncbi.nlm.nih.gov/27919588/

Materials and methods

L70-80 - please provide rationale for those

L82-84 - please provide data why exactly those questionnaires were used, on of the references I brought up earlier might come in handy

L99-116 - there were large amount of different tools used for patients AHI assessment - this should be emphasized in Study limitations section

Discussion

L215-218 - please elaborate why UPPP is not a treatment of choice now, with relevant citation(s) - or remove if redundant

Author Response

Thank you very much for your comments. We have revised our manuscript to address all the issues you pointed out. Please see the attachment

Round 2

Reviewer 2 Report

The authors have done an excellent job of addressing any issues. This is an interesting paper with inventive methodology. I applaud the authors for their great work.